# Optimization Design of Surface Wave Electromagnetic Acoustic Transducers Based on Simulation Analysis and Orthogonal Test Method

**DOI:** 10.3390/s22020524

**Published:** 2022-01-11

**Authors:** Ju Lan, Jingjun Zhang, Xiaojuan Jia, Ruizhen Gao

**Affiliations:** College of Mechanical and Equipment Engineering, Hebei University of Engineering, Handan 056038, China; lanju081212@163.com (J.L.); zhangjingjun@hebeu.edu.cn (J.Z.); gaoruizhen@hebeu.edu.cn (R.G.)

**Keywords:** electromagnetic acoustic transducers, surface waves, 2-D simulation analysis, orthogonal test method, optimal design

## Abstract

The energy conversion of electromagnetic acoustic transducers (EMATs) is typically lower, which seriously restricts the application of EMATs in the field of non-destructive testing and evaluation. In this work, parameters of surface wave EMATs, including structural parameters and electrical parameters, are investigated using the orthogonal test method to improve the transducer’s energy conversion efficiency. Based on the established finite element 2-D model of EMATs, the amplitude of the displacement components at the observation point of a plate is the optimization objective to be maximized with five parameters pertaining to the magnets, meander-line coils, and excitation signal as design variables. Results show that the signal amplitude of EMATs is 3.48 times on in-plane and 3.49 times on out-of-plane, respectively, compared with the original model. Furthermore, a new material (amorphous nanocrystalline material of type 1K107) is applied to optimize the magnetic circuit of EMATs and enhance the eddy current in an aluminum plate to increase the signal amplitude. Finally, the signal amplitudes obtained from the three types of models, that is, the original one, the optimization one after an orthogonal test, and the optimization one with the addition of magnetic concentrators, are analyzed and compared, indicating that the signal amplitude, compared with the original one, is 6.02 times on in-plane and 6.20 times on out-of-plane, respectively.

## 1. Introduction

Electromagnetic Acoustic Transducers (EMATs) are progressively becoming a mainstream technology in the field of Non-Destructive Evaluation (NDE). Compared with conventional piezoelectric transducers, EMATs possess apparent advantages as follows: detection with indirect contact, operation without coupling agent, preferable directivity, high-temperature operation, fast-moving objects detection, and the ability to generate and receive various ultrasonic waves such as shear horizontal (SH) waves, Lamb waves, surface waves, and so on [1,2,3,4,5,6]. In contrast, electromagnetic ultrasonic transducers have some drawbacks that cannot be ignored, such as low intensity of ultrasonic signal excited by EMATs, wide ultrasonic radiation mode, and simultaneously, energy dispersion, which leads to low energy conversion efficiency of EMAT; consequently, the wide application of electromagnetic ultrasonic technology is greatly limited [1,7,8]. For the sake of improving the energy conversion efficiency of EMATs, researchers have carried out extensive studies and investigations.

From the perspective of energy conversion mechanism of EMATs, Huang and Jia et al. focused on the energy conversion principle of EMAT and established the detection process model of EMATs by analytical method and numerical analysis method [9,10]. Kang et al. established the finite element model of the EMAT excitation process of tortuous coil and optimized its design by orthogonal experiment, which improved the energy conversion efficiency of EMATs [11]. Optimization of bias magnetic field: researchers proposed a new magnet arrangement method, in which multiple permanent magnets were symmetrically arranged around the ferromagnetic core, and the magnetic flux density measured on the sample surface exceeded 3T [12]. Dhaylan et al. applied a soft material (${\mathrm{Fe}}_{60}{\mathrm{Ni}}_{10}V_{10}B_{20}$) between the permanent magnet and the coil conductors to increase the strength of the bias field [13]. In view of the structure and arrangement of permanent magnets, Song et al. placed a dual-period permanent magnet electromagnetic ultrasonic transducer (PPM EMATs) symmetrically on opposite positions on both sides of the test specimen and stimulated pure SH1 guided waves for plate inspection [14]. Wu et al. indicated that almost all factors have an obvious impact on the theoretical maximum value of amplitude, while only the magnet-to-coil distance can affect the lift-off performance effectively [15]. Since the excitation of surface wave has a strict corresponding relationship with frequency and coil spacing, it is not suitable for our paper to study the change of coil spacing without considering frequency. Sun et al. proposed a point-focused SH-wave transducer composed of fan-shaped permanent magnets. The transducer with this structure can make the shear-horizontal guided waves generated by the coil realize phase superposition and maximize the amplitude of SH waves during propagation [16].

Electromagnetic ultrasonic surface waves can detect defects on the surface and in close proximity of the surface of materials, which is of great significance for engineering applications. For example, researchers have applied ultrasonic surface wave defects detection [3,17]. Therefore, it is necessary to optimize the design of surface wave EMAT to improve detection efficiency. Tkocz et al. established a surface wave finite element model, which was used to analytically deduce the optimal coil width of EMAT under harmonic excitation, and they put forward the concept of a phase array surface wave to improve the signal amplitude [18]. To detect defects within 10 mm in steel plates, Rachel put forward a new differential coil EMAT, which can measure two components at the same time and analyze in-plane and out-of-plane velocities [19]. Sun et al. proposed a new permanent magnet and coil combination consisting of periodic permanent magnet and return dislocation tortuous coil, which can improve the received signal intensity of a Rayleigh wave [20]. Accordingly, researchers proposed a Rayleigh wave EMAT model with a narrower width of the magnet than the coil and non-uniform distribution of the coil, which can effectively improve the signal intensity of excitation [21]. Lei et al. studied the propagation process of surface wave EMATs on the basis of establishing a 3-D model, and they studied the influence of different parameters on surface waves by using the orthogonal test method and the optimized surface waves amplitude increased by 25.2% [11,22]. The amplitude of electromagnetic ultrasonic vibration is directly related to the Lorentz force [5].

However, detailed research and analysis on how to increase the influence of the Lorentz force are far from sufficient. Therefore, in this paper, an improved two-dimensional surface wave model is proposed to analyze the factors affecting the Lorentz force, which enhances the surface wave signal intensity and improves the detection efficiency. The structure of this paper is as follows: Firstly, a 2-D finite element model is established to study the propagation process of surface waves in an aluminum plate. Secondly, the impact of each parameter on the EMAT amplitude of surface waves is obtained by orthogonal experimentation. The changes of induced eddy current and static magnetic field intensity are compared when different parameter combinations are carried out, and these parameters are optimized to maximize the amplitude of excited surface waves. Finally, based on the orthogonal optimization model, a novel material (amorphous nanocrystalline material of type 1K107) is applied to optimize the magnetic circuit, the Lorentz force density in the *X*-direction and *Y*-direction and the eddy current density on the surface of an aluminum plate. The results of these three aspects show that ANSM can effectively improve the performance of surface wave EMAT, which is consistent with the original purpose of this paper; that is, it can enhance the strength of the surface wave signal. 

## 2. Transduction Mechanism and Governing Equation of EMATs

Generally speaking, the electroacoustic transduction of EMATs involves three energy transfer mechanisms: the Lorentz force, the magnetostrictive force, and the magnetization force [23]. In contrast, the specimen built in this paper is an aluminum plate, which belongs to a nonferromagnetic material, so that the Lorentz force mechanism is the sole mechanism of the electroacoustic transducer. Surface waves EMATs are generally composed of a permanent magnet, a meander-line-coil, and the specimen. By optimizing the parameters of these components, the detection effectiveness of surface wave EMATs can be significantly improved.

In the process of surface wave excitation, a high-frequency alternating current $J_{c}$ is charged in the coil, an alternating electric field is generated around the meander-line-coil, and the alternating electric field produces induced current on the specimen. The induced vortices generate Lorentz forces $F_{L}$ inside the aluminum plate under the action of static bias magnetic field $B_{s}$ provided by a permanent magnet. Depending on the knowledge of the elastic dynamics, it will cause elastic deformation of the specimen and periodic particle vibration of the particles. When the vibration propagates in the form of waves, ultrasonic waves form. The total intensity of the magnetic field is the superposition of alternating magnetic field and permanent magnet stationary magnetic field [6]. Figure 1 illustrates the generation process of surface waves based on the Lorentz force.

The excitation progress of ultrasonic waves based on Lorentz force can be described as the follow equations: (1)$$\nabla \times H_{d}=J_{c}$$
(2)$$B_{d}=\mu_{m}H_{d}$$
(3)$$\nabla \times E_{e}=-\partial B_{d}$$
(4)$$J_{e}=\gamma E_{e}$$
(5)$$F_{L}=F_{d}+F_{s}=J_{e}\times B_{d}+J_{e}\times B_{s}$$

In Formula (1), $H_{d}$ is the magnetic field intensity of the alternating magnetic field; In Formula (2), $B_{d}$ is the magnetic induction intensity of the alternating magnetic field; $\mu_{m}$ is the relative permeability of aluminum plate. In Formula (3), $E_{e}$ is the electric field intensity generated by the alternating magnetic field induction; γ is the electrical conductivity of aluminum plate. In Formula (4), $J_{e}$ is the eddy current density in the aluminum plate, and in Formula (5), $F_{L}$ is the Lorentz force in a region of the aluminum plate, which is equal to the vector sum of static Lorentz force $F_{s}$ and dynamic Lorentz force $F_{d}$. Obviously, the Lorentz force $F_{L}$ is closely related to the eddy current density, magnetic induction intensity of alternating magnetic field, and static magnetic field intensity. Increasing the Lorentz force can make the particle vibration amplitude larger; hence, the detection efficiency of the electromagnetic ultrasonic transducer will be improved distinctly.

## 3. Surface Wave EMAT Modeling and Simulation Analysis

In this paper, the two-dimensional aluminum plate surface wave EMAT is analyzed using COMSOL Multiphysics. The built two-dimensional finite element model is illustrated in Figure 2, consisting of a coil, permanent magnet, specimen, and air field, where the air field is used to simulate the space in the electromagnetic ultrasonic working environment. To reduce the influence of reflection on signal analysis, the left and right boundaries of the aluminum plate are set as low reflection boundaries in this model. 

The mesh size of the aluminum plate was set to be less than 1/8 of the surface wave length to ensure the solution accuracy and shorten the solution time as much as possible; therefore, it was set to 0.7 mm. As the surface wave energy is mainly concentrated in the area of twice the wavelength, the area beneath the surface of the aluminum plate is refined with the maximum mesh size set as 0.2 mm.

Table 1 shows the geometric parameters of surface wave EMATs and some physical properties of the coil, permanent magnet, and aluminum plate in a finite element model. In this paper, the Nd-Fe-B permanent magnet with high remanence is utilized to provide a bias magnetic field. The coil has eight turns of conductors, and the conductors are evenly spaced. According to the propagation of the surface wave in aluminum plate being about 2930 m/s, the surface wave length λ is calculated to be 5.86 mm according to the formula λ = c/*f*, where c is the theoretical velocity of surface wave propagation in aluminum and *f* is the frequency of the excitation current $J_{c}.\;$ In order to obtain constructive interference of the surface wave that EMAT generates, so as to acquire ultrasonic surface waves with larger displacement amplitude and stronger directivity, the center distance L of adjacent coils should be made half the length of the wavelength of the surface wave, that is, L = 2.93 mm, and the distance between the coil and the aluminum plate is 0.1 mm.

After grid division and time domain calculation, the vector and distribution diagrams of in-plane and out-of-plane displacements of surface waves in an aluminum plate at different times can be obtained, as shown in Figure 3. In this paper, u represents in-plane displacement and v represents out-of-plane displacement, and point P (100, –0.5) is selected for analysis.

Figure 3 shows that the meander-line-coil EMAT can generate surface waves propagating along the surface and near the surface of an aluminum plate, along with bulk waves. As the propagation time increases, the bulk waves decay rapidly, while the surface waves decay slowly. Figure 4 shows the in-plane and out-of-plane displacements at point P, the time difference of the maximum out-of-plane displacement between the direct surface wave and the interface reflection is 41.4 µs, and the distance between the two wave packets is 120 mm, the calculated wave propagation velocity is 2898.6 m/s, and the error value is 1.07% with the theoretical surface wave propagation velocity, which proves that the model has relatively high accuracy.

## 4. Optimal Design of EMATs

### 4.1. Orthogonal Test Design

Researchers have studied the influence of single factors such as the width and thickness of the coil, permanent magnet height, and coil lift distance on the performance of EMATs that generate ultrasonic wave; however, the research on the influence of combination parameters on EMAT is relatively insufficient [1,8]. For the sake of obtaining the optimal transducer parameter combinations with the best performance of surface wave utilized by meander-line-coil EMAT, the essential factors affecting the excitation performance are selected based on the established model, and the influence of each parameter on the excitation performance is analyzed as the basis of optimization. On account of a large amount of calculation and a mass of parameter combinations of the finite element model, the orthogonal experimental method is appropriate to be employed to study the optimization of EMAT excited surface waves. The orthogonal test method provides an efficient and systematic way to determine the influence of the various EMAT parameters and determine the preferred values of the parameters so that an optimal result can be found with only a few numerical experiments.

In this paper, surface wave displacements and eddy current density $J_{e}$ are taken into consideration as research objects, and EMAT parameters such as the width and thickness of the coil and permanent magnet height and so on are taken as influence parameters to design an orthogonal experiment.

The EMAT parameters under consideration include coil width $w_{1}$ and thickness $t_{1},$ lift-off distance $h_{1},$ magnet height $t_{2},$ and alternating current peak value $I_{t},$ Interestingly, the effects of varying the lift-off distance $h_{1}$ and the excitation current $I_{t}$ have been studied in depth [8]. The orthogonal test in this paper mainly studies the influence on eddy current density, alternating magnetic induction intensity, and surface wave displacement by simulating various combinations of parameters. 

The ranges of variation of the EMAT parameters, which are the factors of the array, are chosen according to the common specification and manufacturing process of EMAT in aluminum plate detection [11], which are: $w_{1}$: 0.2–0.8 mm, $t_{1}$: 0.5–1.5 mm, $h_{1}$: 0.1–0.7 mm, $t_{2}$: 20–35 mm, $I_{t}$: 50–200 A. These five parameters are divided into four levels, as shown in Table 2, and an orthogonal test is carried out, using the orthogonal array $L_{16}$ ($4^{5}$) [11], as shown in Table 3. According to the 16 groups of data in Table 3, the finite element model was changed respectively. Following simulation of the calculation, $u_{m}$ (amplitude of in–plane displacement) and $v_{m}$ (amplitude of out–of–plane displacement) were obtained under 16 conditions respectively, as shown in the last two columns in Table 3.

### 4.2. Analysis of Orthogonal Test Results

The influence of each level of factor was calculated by counting the arithmetic average $k_{i}$ of $u_{m}$ and $v_{m}$ of each factor at the same level according to the orthogonal experiment results (*i* stands for the serial number of each level of factor, *i* = 1, 2, 3, 4) [22], and the value of *R* (*R* denotes the difference between the largest and smallest values of $k_{i}$) of each factor was calculated according to $k_{i}$, as shown in Table 4. It can be seen from Table 4 that factors with the largest range *R* have the most significant impact on $u_{m}$ and $v_{m}$, which is the essential factor affecting the excitation of surface waves. The sequence of influence degree of each factor are as follows: $I_{t}$ > $h_{1}$ > $w_{1}$ > $t_{2}$ > $t_{1}.$ It is noteworthy that the influence trend of these factors on $u_{m}$ is consistent with $v_{m}$. According to the relationship between the level taken by each factor and $k_{i}\;$ that corresponds to each factor in Table 4, the trend graph of the influence of each factor on $u_{m}$ and $v_{m}$ is shown in Figure 5 and Figure 6.

It can be seen from Table 3, Figure 5 and Figure 6 that the most significant factors affecting the performance of surface waves generated by EMAT are excitation current peak value $I_{t}$ and coil lift-off distance $h_{1}.$ Increasing $I_{t}$ or decreasing $h_{1}$ can improve the displacement amplitude of surface waves significantly. The coil width and permanent magnet height are the second most important factors. Appropriately increasing $w_{1}$ and $t_{2}$ can enhance the excitation performance of EMAT. The thickness of the coil is least affected by the way. Therefore, for the sake of improving the excitation performance of EMAT, the optimal parameter combination is as follows: $w_{1}$ = 0.4 mm, $t_{1}$ = 0.06 mm, $h_{1}$ = 0.1 mm, $t_{2}$ = 30 mm, $I_{t}$ = 200 A. This set of parameters is not included in the orthogonal array. Simulation of the EMAT with these optimal parameters shows that in the model, $u_{m}=17.55\times 10^{-8}$ mm, $v_{m}=7.32\times 10^{-7}\;\mathrm{mm}$, which is larger than any value of $u_{m}$ and $v_{m}$ in Table 2, the value of the optimal parameter combination is 4.78 times and 4.55 times that before orthogonal optimization; thus, the effectiveness and efficiency of the orthogonal experiment are proved again. The comparison of the original and orthogonal surface wave displacements is shown in Figure 7.

### 4.3. Optimal Design

Based on the orthogonal optimization model, this paper introduces a new material, which can not only increase the magnetic induction intensity but also enhance the eddy current generated in an aluminum plate so as to optimize the surface wave EMATs magnetic circuit and realize the enhancement of ultrasonic surface wave signals. Amorphous nanocrystalline material of type 1K107 (hereinafter referred to as ANSM) has many other prominent advantages: high initial permeability, sound high-frequency performance, low loss and coercive force, and excellent saturation magnetic induction. On the strength of the ANSM material, it is made into rectangular lamination with a certain thickness, which covers the upper surface of the meander-line-coil to reinforce the eddy current on the surface of an aluminum plate, so as to optimize the magnetic circuit to enhance the surface wave signals.

The amorphous nanocrystalline material of type 1K107 is applied in this paper. The thickness of each piece is 0.02 mm, the conductivity is $1.25\times 10^{6}\;S/m$, the relative dielectric constant is 1, the relative permeability is 20,000, and the saturation magnetic induction intensity is 1.25 T. A slice of ANSM with a thickness of 0.1 mm was produced and placed between the coil and the permanent magnet, and for better effect, the slice is put closely to the upper surface of the coil, as shown in Figure 8.

After simulation, the amplitudes of in-plane and out-of-plane displacements of surface wave are as follows: $u_{m}=21.90\times 10^{-8}\;\mathrm{mm}$, $v_{m}=7.32\times 10^{-7}\;\mathrm{mm}$; compared with the model without the new material, the displacement amplitude is obviously increased, which proves the effectiveness of the ANSM. The influence of the thickness of the new material on the excitation performance of surface wave EMATs was studied by maintaining the other EMAT parameters unchanged. The thickness of ANSM was denoted as $t_{3},$ the levels of $t_{3}$ are 0.2, 0.4, and 0.6 mm, and based on the orthogonal optimization model, the three models with different thickness levels are emulated respectively. The results are presented in Figure 9. 

It can be seen from Figure 9 that the surface wave displacement increases linearly with the increase in $t_{3}$. The reason for this result is that by covering the coil with a new material, the magnetic circuit is optimized, and the eddy current on the surface of the aluminum plate is greatly increased, thus enhancing the vibration displacement of the surface waves. Considering that the production process of this material is not complicated, easy to obtain, and cheap, this paper chooses 0.6 mm thickness material as the final optimization model.

Figure 10 shows the comparison of flux density modes of static magnetic field before and after the new material is added. In Figure 10, the magnetic induction intensity at the left and right edges of the permanent magnet is the largest, while the magnetic induction intensity in the middle is weak, and the magnetic field is evenly distributed. Before adding the new material of the static magnetic field, the magnetic flux density modulus maximum value can reach 1.74 T, and after adding the new material, the maximum static magnetic field of the magnetic flux density model is 2.01 T; the reason for the phenomenon is that the relative permeability of ANSM is relatively greater than the air permeability. In other words, after adding ANSM in the EMAT model, the permanent magnet provides a much smaller offset magnetic field for meander-line-coils and an aluminum plate.

Figure 11 displays the comparison of components of Lorentz force density on an aluminum plate surface in *X* and *Y*-directions at t = 6 µs before and after adding ANSM. It can be seen that the Lorentz force density in the *X*-direction is symmetric about the center point, and the Lorentz force on the surface of the aluminum plate directly below the adjacent coil is opposite. The maximum value is about $2.52\times 10^{7}\;N/m^{3}$ before adding ANSM and about $4.68\times 10^{7}\;N/m^{3}$ after adding ANSM. The Lorentz force density in the *Y*-direction is symmetrical along the center line, with the same direction and relatively uniform distribution. The maximum value is about $1.45\times 10^{8}\;N/m^{3}$ before adding ANSM and $1.61\times 10^{8}\;N/m^{3}$ after applying ANSM. Obviously, after adding ANSM, the Lorentz force increases significantly in both X and Y-directions, and the increase in the *X*-direction is larger than that in the Y direction.

The eddy current distribution diagram under the adjacent wires in the aluminum plate at *t* = 39.25 µs is obtained, as shown in Figure 12 and Figure 13. As can be seen in Figure 12, the eddy current beneath the adjacent conductor is equal in value and opposite in direction. It can be seen that after the addition of the new material, the maximum eddy current in the aluminum plate directly under the coil increases from 28.2 A/$m^{2}$ to 167A/$m^{2},$ which indicates that the eddy current of the aluminum plate increases significantly.

Figure 14 shows the comparison of the surface wave displacement amplitude between the original model, the orthogonal optimization model, and the model with the addition of the ANSM material. Apparently, the displacement of the model with the ANSM material increases significantly; the signal amplitude, compared with the original one, is 6.02 times on in-plane and 6.20 times on out-of-plane respectively, which proves the effectiveness and accuracy of this material.

In general, it can be seen from the comparative analysis that the static flux density, the X and Y components of the Lorentz force density, and the eddy current on the surface of the aluminum plate are significantly increased after the addition of ANSM, which verifies the effectiveness of ANSM in increasing the amplitude of surface waves.

## 5. Conclusions

Aiming at enhancing the ultrasonic signal intensity of the surface wave EMATs for the inspection of an aluminum plate, an optimal design method is carried out by using the orthogonal test method based on the two-dimensional model previously established. The influence of various parameters on the amplitude is analyzed and compared. Experiments indicate that the signal amplitude of the model after orthogonal experiments is 3.48 times in the *X*-direction and 3.49 times in the *Y*-direction compared with the original model. More importantly, on the basis of the model added with new material after the orthogonal experiment, the signal amplitude, compared with the original one, is 6.02 times on in-plane and 6.20 times on out-of-plane respectively, which verifies the validity of the proposed method. In the data variation ranges of the EMAT parameters, conclusions are drawn as follows:(1)Although the lift-off distance and excitation current play the most significant role in the enhancement of signal amplitude of surface waves, the proper choice of EMAT parameters, such as the thickness of the magnet and coil as well as the width of coil conductors can significantly increase the signal amplitude.(2)The amplitude of surface wave displacement increases linearly with the increasing excitation current or decreasing lift-off distance, and these two factors have the most significant effect on surface wave displacement. The effect of conductor width and permanent magnet height on surface waves cannot be ignored. Appropriately increasing the wire width and permanent magnet height can also increase the displacement amplitude. The coil thickness has little influence on the displacement, so the conductor thickness should be as small as possible under the condition that the manufacturing process allows.(3)The lift-off distance and excitation current have similar magnitude effects, and they have a much stronger influence on the signal amplitude of the surface waves excited by the Lorentz forces. This allows the properties of the excited surface waves to be adjusted to meet the requirements of actual applications.(4)Covering the meander-line-coil with a new material ANSM can dramatically optimize the magnetic circuit, enhance the eddy current density indicated on aluminum plate, improve the Lorentz force density in the *X*-direction and *Y*-direction, and also increase the dynamic magnetic induction intensity, thus reinforcing the signal amplitude of ultrasonic surface waves. Simply put, the new material ANSM can improve the overall performance of surface wave EMATs.

The significance of this paper is that the signal strength can be significantly improved by covering ANSM material of a certain thickness above the coil without a complex structure, thus achieving the purpose of convenient detection and improving detection efficiency, in addition, providing a convenient method for future practical industrial detection.

## Figures and Tables

**Figure 1 sensors-22-00524-f001:**
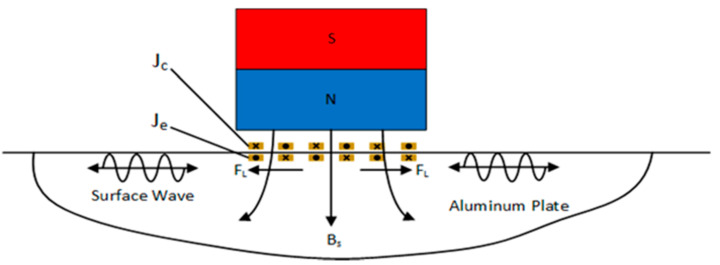
Schematic diagram of the ultrasonic generation mechanism of a meander-line-coil surface wave EMAT based on the Lorentz force.

**Figure 2 sensors-22-00524-f002:**
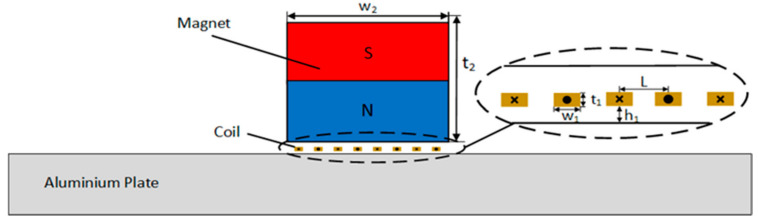
Two-dimensional (2-D) physical model of the surface wave EMAT.

**Figure 3 sensors-22-00524-f003:**
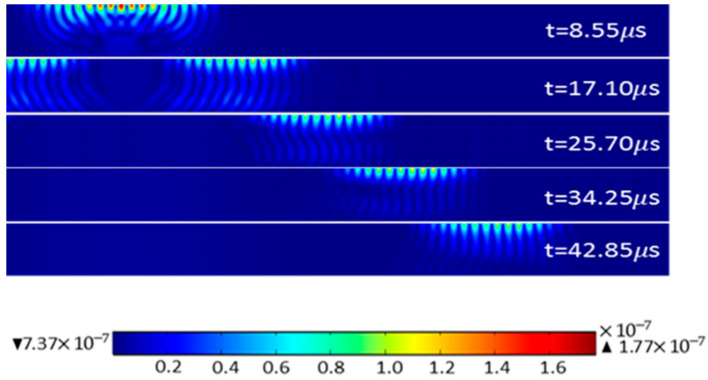
Displacement distribution of surface waves.

**Figure 4 sensors-22-00524-f004:**
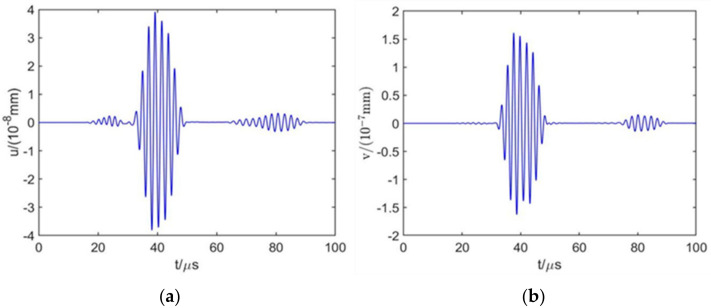
Surface wave displacements at point P. (**a**) In-plane displacement; (**b**) out-of-plane displacement.

**Figure 5 sensors-22-00524-f005:**
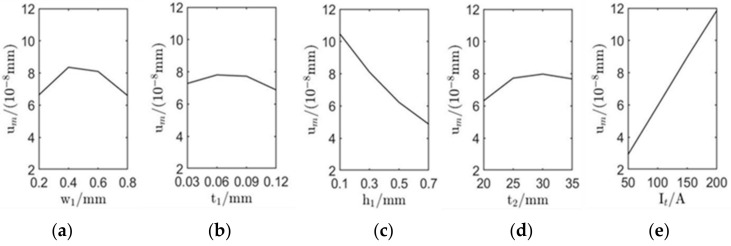
Influence of varying the EMAT design parameters on the amplitude of in-plane displacement of surface waves: (**a**–**e**) successively show the influence of coil width $w_{1},$ coil thickness $t_{1},$ coil lift distance $h_{1},$ permanent magnet thickness $t_{2}$, and excitation current peak value $I_{t}$ on the amplitude of in-plane displacement.

**Figure 6 sensors-22-00524-f006:**
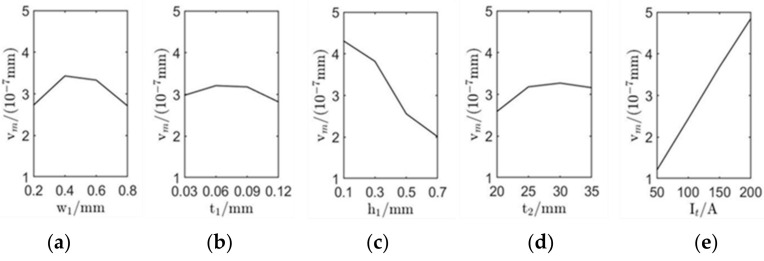
Influence of varying the EMAT design parameters on the amplitude of out-of-plane displacement of surface waves: (**a**–**e**) successively show the influence of coil width $w_{1},$ coil thickness $t_{1},$ coil lift distance $h_{1},$ permanent magnet thickness $t_{2}$, and excitation current peak value $I_{t}$ on the amplitude of out-of-plane displacement.

**Figure 7 sensors-22-00524-f007:**
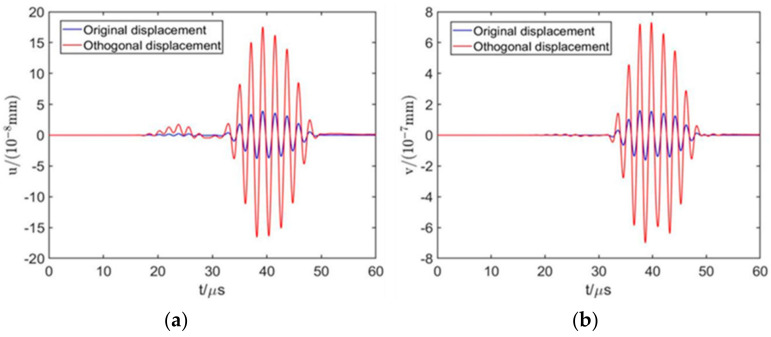
Comparison of surface waves between original and orthogonal EMATs. (**a**) In-plane displacement and (**b**) out-of-plane displacement.

**Figure 8 sensors-22-00524-f008:**
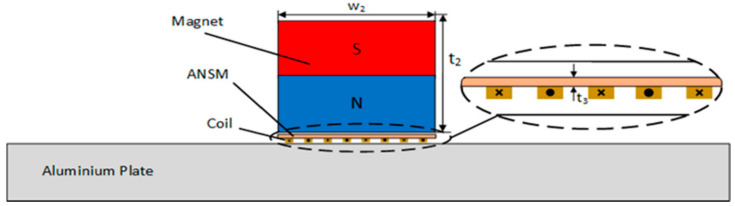
Schematic diagram of the finite element model for surface wave EMATs after adding ANSM.

**Figure 9 sensors-22-00524-f009:**
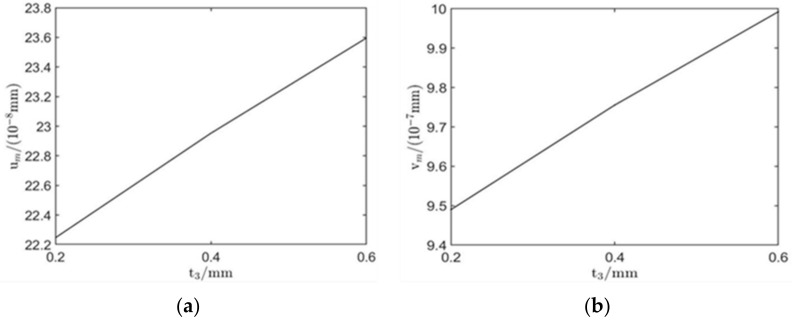
The influence on surface wave amplitude by changing the thickness. (**a**) Amplitude of in-plane displacement and (**b**) amplitude of out-of-plane displacement.

**Figure 10 sensors-22-00524-f010:**
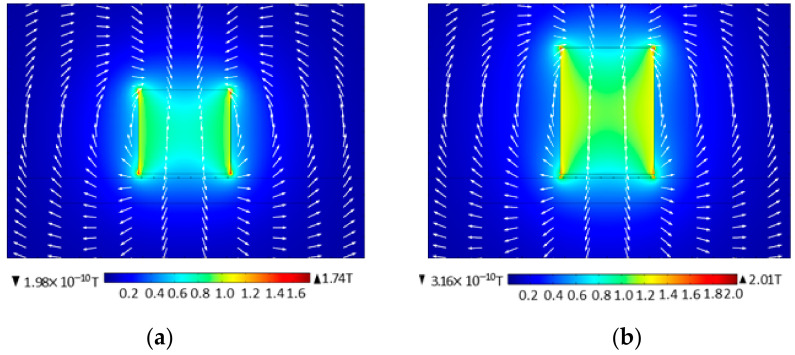
Flux density mode of static magnetic field before and after applying ANSM. (**a**) Before adding ANSM and (**b**) after adding ANSM.

**Figure 11 sensors-22-00524-f011:**
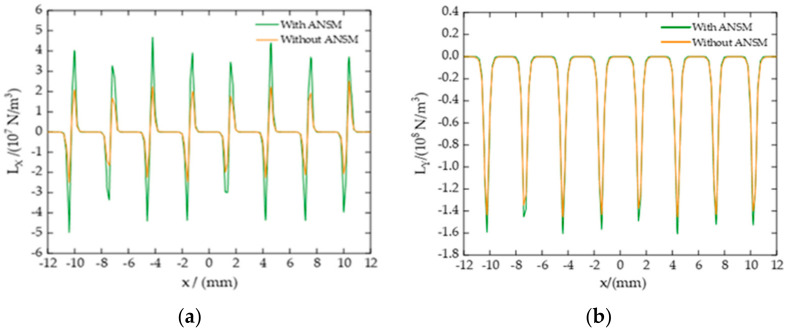
The Lorentz force density on the surface of an aluminum plate at *t* = 6 µs. (**a**) The Lorentz force density on an aluminum plate surface in the *X*-direction and (**b**) the Lorentz force density on aluminum plate surface in the *Y*-direction. Based on the orthogonal optimization model, the green line represents the Lorentz force density after ANSM is applied in the model, and the yellow curve represents the Lorentz force density without ANSM.

**Figure 12 sensors-22-00524-f012:**
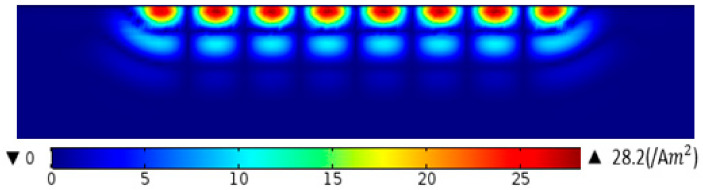
Eddy current distribution in aluminum plate before adding ANSM.

**Figure 13 sensors-22-00524-f013:**
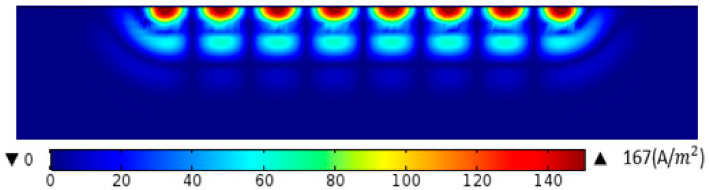
Eddy current distribution in aluminum plate after adding ANSM.

**Figure 14 sensors-22-00524-f014:**
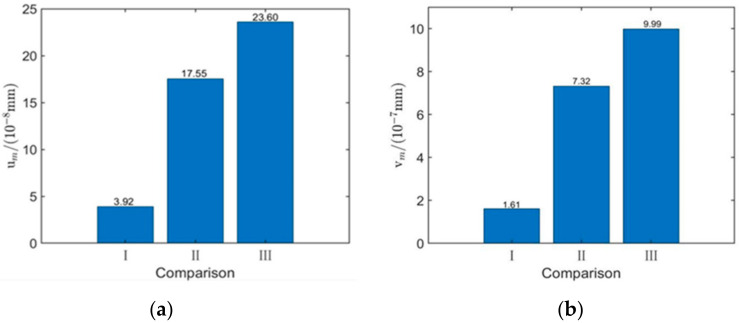
Comparison of amplitude between original, orthogonal, and optimal displacement. (**a**) Amplitude of in-plane displacement and (**b**) amplitude of out-of-plane displacement.

**Table 1 sensors-22-00524-t001:** Parameters and values of surface wave EMATs in the finite element model.

Object	Parameter	Value
Coil	Width	0.1 mm
	Thickness	0.05 mm
	List-off distance	0.1 mm
	Resistivity	$1.7\times 10^{-8}$ Ωm
Magnet	Width	22 mm
	Thickness	20 mm
	Magnetic flux density	1.35 T
	List-off distance	1.0 mm
Aluminum Plate	Width	320 mm
	Thickness	20 mm
	Density	2700 kg/m^3^
	Electrical conductivity	$3.77\times 10^{7}$ S/m
	Young’s modulus	$70\times 10^{9}$ Pa
	Poisson’s ratio	0.33
Excitation Current	Peak value	50 A
	Frequency	500 kHz

**Table 2 sensors-22-00524-t002:** Ranges of variation of EMAT parameters.

Levels	$w_{1}\;(\mathrm{mm})$	$t_{1}\;(\mathrm{mm})$	$h_{1}\;(\mathrm{mm})$	$t_{2}\;(\mathrm{mm})$	$I_{t}\;(A)$
1	0.2	0.03	0.1	20	50
2	0.4	0.06	0.3	25	100
3	0.6	0.09	0.5	30	150
4	0.8	0.12	0.7	35	200

**Table 3 sensors-22-00524-t003:** Ranges of variation of EMAT parameters.

Run	$w_{1}\;$	$t_{1}\;$	$h_{1}\;$	$t_{2}\;$	$I_{t}\;$	$u_{m}$	$v_{m}$
	(mm)	(mm)	(mm)	(mm)	(A)	$\left(10^{-8}\;\mathrm{mm}\right)$	$\left(10^{-7}\;\mathrm{mm}\right)$
1	0.2	0.03	0.1	20	50	3.95	1.63
2	0.2	0.06	0.3	25	100	6.53	2.69
3	0.2	0.09	0.5	30	150	7.85	3.23
4	0.2	0.12	0.7	35	200	8.23	3.37
5	0.4	0.03	0.3	30	200	13.9	5.67
6	0.4	0.06	0.1	35	150	13.6	5.60
7	0.4	0.09	0.7	20	100	3.52	1.46
8	0.4	0.12	0.5	25	50	2.43	1.01
9	0.6	0.03	0.5	35	100	5.53	2.28
10	0.6	0.06	0.7	30	50	2.00	0.83
11	0.6	0.09	0.1	25	200	16.2	6.65
12	0.6	0.12	0.3	20	150	8.68	3.56
13	0.8	0.03	0.7	25	150	5.76	2.37
14	0.8	0.06	0.5	20	200	9.12	3.73
15	0.8	0.09	0.3	35	50	3.35	1.39
16	0.8	0.12	0.1	30	100	8.17	3.35

**Table 4 sensors-22-00524-t004:** Analysis of orthogonal test results.

Amplitude		$w_{1}\;$	$t_{1}\;$	$h_{1}\;$	$t_{2}\;$	$I_{t}\;$
		(mm)	(mm)	(mm)	(mm)	(A)
	*k* _1_	6.64	7.28	10.48	6.32	2.93
	*k* _2_	8.36	7.81	8.11	7.73	5.94
$u_{m}$	*k* _3_	8.10	7.73	6.23	7.98	8.97
$(10^{-8}$ mm)	*k* _4_	6.60	6.88	4.88	7.68	11.9
	*R_u_*	1.76	0.93	5.60	1.66	8.97
$\mathrm{Influence}\;\mathrm{Rank}\;I_{t}$$(8.97)\;>\;h_{1\;}$$(5.60)\;>\;w_{1}$$(1.76)\;>\;t_{2}$$(1.66)\;>\;t_{1}$(0.93)						
	*k* _1_	2.73	2.98	4.31	2.59	1.21
	*k* _2_	3.43	3.21	3.82	3.18	2.44
$v_{m}$	*k* _3_	3.33	3.18	2.56	3.27	3.69
$(10^{-7}$ mm)	*k* _4_	2.71	2.82	2.01	3.16	4.85
	*R_v_*	0.72	0.39	2.30	0.68	3.64
$\mathrm{Influence}\;\mathrm{Rank}\;I_{t}$$(3.64)\;>\;h_{1}$$(2.30)\;>\;w_{1}$$(0.72)\;>\;t_{2}$$(0.68)\;>\;t_{1\;}$(0.39)						

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
