# Peer review of "Optimization Design of Surface Wave Electromagnetic Acoustic Transducers Based on Simulation Analysis and Orthogonal Test Method"

_sensors, 2022, doi:10.3390/s22020524_

Round 1

Reviewer 1 Report

This paper presents a surface wave optimization method using orthogonal test. In general, the structure of this paper is clear, and it has innovations. The reviewer gives the following comments for the author's reference:
1. This paper seems to lack experimental verification, it is recommended to supplement the experiment because it is very important.
2. There are many existing studies (including keywords: orthogonal test, acoustic waves, optimization, ndt, simulation), including the author's own papers Optimal Design of Point–Focusing Shear Vertical Wave Electromagnetic Ultrasonic Transducers Based on Orthogonal Test Method. This seems to just change the type of wave and use a similar method. The author can emphasize the innovations in the paper's Abstract.
3. Refer to comment-2, some related papers should be compared in the introduction. Such as ''Orthogonal Optimal Design Method for Spiral Coil EMAT Considering Lift-off Effect: A Case Study", and ''Point-Focusing Shear-Horizontal Guided Wave EMAT Optimization Method Using Orthogonal Test Theory".
4. How the orthogonal experiment parameters are selected needs more explanation.

Reviewer 2 Report

The article discusses a simulation study using COMSOL software providing an optimized design of EMAT devices by enhancing design parameters to improve the amplitude of acoustic waves. parametric study of design parameters are discussed and a proposed material is presented as a potential novelty to enhance the eddy current  and the magnetic circuit. 

The paper structure is appropriate and the figures support the mentioned idea. Introduction provides sufficient background information with relevant bibliography. The study has a good organization and flow of ideas that would make it easily understandable for the average reader.

I recommend the manuscript to be published in the current form.

Reviewer 3 Report

The propagation process of surface wave EAMTs is studied in the paper. For improving the amplitude of acoustic waves, the influence of various parameters on surface waves is investigated by the orthogonal test method, and the optimal parameter combination is obtained.

The topic is very interesting. The manuscript has some of its technical merits. The structure of the manuscript is also well. It should fall into Journal Sensors. However, several questions might pay attention to:

  1. The manuscript’s language does not reach the level of publishing in an international journal. One professional English native writer should polish it.
  2. In section 1, the authors listed many references in the text. However, the authors haven’t highlighted the significance of your work. Recommend the author should emphasize your research significance as soon as possible.
  3. In part 3, section 3, the authors described the parameters of your simulation model. If one table was provided to explain the simulation model, it would be clearer.
  4. In Figures 5 & 6, the authors ignored the subtitle of each sub-figure. Recommend the authors should provide it.
  5. There is only a computer simulation experiment and no real physical experiment to validate your statement in the manuscript. The authors need to address it.
  6. The discussion is very simple. The authors have not provided a comparison of enough previous literature. Recommend authors should extend your discussion.
  7. The authors can improve your conclusions. It might also suggest future research.

Hopefully, this will help in the revision of the manuscript.

Reviewer 4 Report

The article deals with the issue of the increase of amplitude in electromagnetic acoustic transducers via FEM analysis and proper coating. According to performed analysis, the amplitude was successfully significantly increased, however, this result was not verified in the physical experiment.

The article is very interesting, but some mistakes were appear:
- are all affiliations of the authors (1-4) the same?
- dot is missing after name initial in lines 60 (Tkocz J.), 63 (Rachel S.E.), 65 (Sun H.), 70 (Lei K.) and in the last sentence in line 329.
- you probably want to write in-plane and out-of-plane in line 63, not in-plane and in-plane.
- there is a missing explanation of parameters Jc , Bs , Fd and Fs in Fig. 1 and Eq. 5.
- please, check equation (1) - can it be an equation without "="?
- you mentioned the value of sound velocity in Al plate as 2390 m/s in line 146, and as 2898,6 m/s in line 172. Could you explain the difference?
- you should explain the formula in lines 147-148. Is "c" the velocity? Is "f" the frequency?
- I believe the word "amplitude" in line 195 is not proper.
- I am not sure about the thickness of the coating - you mentioned a value of 0.02 mm in line 266, 0.1 mm in line 268, and 0.2, 0.4 and 0.6 in line 280. Could you explain it? Is it some sort of sandwich structure?
- the effect of ANSM in Fig. 10 could be more obvious if there will be set the same scale. Similarly in Fig. 11 and 12.

Round 2

Reviewer 1 Report

The paper revised well and I would like to accept it.

Reviewer 3 Report

The authors have revised their manuscript according to most of the points.